# The Oncogenic Signaling Pathways in *BRAF*-Mutant Melanoma Cells Are Modulated by Naphthalene Diimide-Like G-Quadruplex Ligands

**DOI:** 10.3390/cells8101274

**Published:** 2019-10-18

**Authors:** Marta Recagni, Martina Tassinari, Filippo Doria, Graziella Cimino-Reale, Nadia Zaffaroni, Mauro Freccero, Marco Folini, Sara N. Richter

**Affiliations:** 1Department of Applied Research and Technological Development, Fondazione IRCCS Istituto Nazionale dei Tumori di Milano, Via G.A. Amadeo 42, 20133 Milan, Italy; marta.recagni@istitutotumori.mi.it (M.R.); g.ciminoreale@virgilio.it (G.C.-R.); nadia.zaffaroni@istitutotumori.mi.it (N.Z.); 2Department of Molecular Medicine, University of Padua, via A. Gabelli 63, 35121 Padua, Italy; martina.tassinari@unipd.it; 3Department of Chemistry, University of Pavia, v. le Taramelli 10, 27100 Pavia, Italy; filippo.doria@unipv.it (F.D.); mauro.freccero@unipv.it (M.F.)

**Keywords:** *BCL-2*, G-quadruplex, *KIT*, melanoma, naphthalene diimides, oncogene promoter

## Abstract

Melanoma is the most aggressive and deadly type of skin cancer. Despite the advent of targeted therapies directed against specific oncogene mutations, melanoma remains a tumor that is very difficult to treat, and ultimately remains incurable. In the past two decades, stabilization of the non-canonical nucleic acid G-quadruplex structures within oncogene promoters has stood out as a promising approach to interfere with oncogenic signaling pathways in cancer cells, paving the way toward the development of G-quadruplex ligands as antitumor drugs. Here, we present the synthesis and screening of a library of differently functionalized core-extended naphthalene diimides for their activity against the *BRAFV600E*-mutant melanoma cell line. The most promising compound was able to stabilize G-quadruplexes that formed in the promoter regions of two target genes relevant to melanoma, *KIT* and *BCL-2*. This activity led to the suppression of protein expression and thus to interference with oncogenic signaling pathways involved in *BRAF*-mutant melanoma cell survival, apoptosis, and resistance to drugs. This G-quadruplex ligand thus represents a suitable candidate for the development of melanoma treatment options based on a new mechanism of action and could reveal particular significance in the context of resistance to targeted therapies of *BRAF*-mutant melanoma cells.

## 1. Introduction

Melanoma, a malignancy that develops from melanin-producing cells, is the most aggressive and deadly type of skin cancer. Though it is a rare tumor, accounting for only 4% of all skin cancers, it is responsible for the highest number of skin cancer-related deaths worldwide and it is the most difficult cancer to treat [1,2]. Chemotherapy is the standard curative option for advanced stage melanoma patients, even though it has an overall low efficacy and limited response rates [1].

The characterization of the melanoma mutational landscape has led to the identification of specific gene/pathway alterations that have had remarkable implications on the management of this disease [3]. In this context, a major breakthrough in the understanding of melanoma biology was the identification of the *BRAFV600E* mutation [3], which contributed to the development of anti-oncogenic compounds targeting activated mutant *BRAF*. Indeed, up to 50% of melanomas derived from the skin without chronic sun damage bear mutations in the *BRAF* gene, with *BRAFV600E* representing up to 90% of the activating mutations [4]. This scenario led to the development of the ATP-competitive *BRAFV600E* selective inhibitor, vemurafenib [5]. This drug has shown dramatic success in clinical trials, as it promotes tumor regression and increases the overall survival of metastatic melanoma patients [6]. Unfortunately, resistance mechanisms that render melanoma unresponsive to other therapeutic strategies develop within a few months from the beginning of treatment [6]. The second most common mutation in melanoma affects the *NRAS* gene in codon 61, producing such mutations as *Q61R* or *Q61K* [3]. These occur in 20–30% of melanoma patients and are mutually exclusive with *BRAF* mutations [1], except in resistant melanomas after targeted therapy, which may harbor co-occurring *BRAF* and *NRAS* mutations [3].

Recent evidence has indicated that the transmembrane receptor tyrosine kinase c-KIT may also be an attractive therapeutic target in melanoma [7]. Genetic alterations of *KIT* in melanoma include somatic gain-of-function mutations and copy number increases of wild-type *KIT* [7], whereas mutant receptors were found only in 2% of all cutaneous melanomas, thus representing a rare event for targeted treatment, and in up to 20% of mucosal, acral, and chronic sun-damaged skin melanomas [8]. A series of different *KIT* mutations, among which *L576P* was detected in one-third of all cases, was found, although many of them are not suitable targets [4].

The identification of druggable mutation-specific oncogene targets significantly contributed to the expansion of the arsenal of available therapies for patients with advanced melanoma over the past few years. The introduction of targeted therapies, such as BRAF (vemurafenib and dabrafenib) and MEK (trametinib and cobimetinib) inhibitors, as single agents or in combination [1,2], led to both improved response rates and mean overall survival of metastatic melanoma patients bearing the *BRAFV600E* mutation or mutant *NRAS* [3,8]. On the other hand, mutant c-KIT may be able to be targeted by tyrosine kinase receptor inhibitors (e.g., imatinib, sunitinib, and dasatinib), although, at present, clinical benefits have been reported only for imatinib in melanoma patients with *KIT* point mutations in exon 11 or 13, and not in those harboring gene amplification [8].

As per any oncogene-targeted therapy, treatment failure is associated with mechanisms of acquired drug resistance, which may rely on the reactivation of MAPK signaling, the activation of substitutive oncogenic pathways, such as that mediated by PI3K/AKT, as well as on the over-activation of growth factor receptors and the capability to evade apoptosis [1,8,9]. In this context, the deregulation of the BCL-2 family of proteins plays a major role in the evasion of melanoma cell apoptosis in response to treatment [9]. Notably, many BCL-2 proteins are downstream factors of the RAS/BRAF/MAPK and PI3K/AKT signaling pathways, the activation of which contributes to the relapse of melanoma from treatment with targeted therapies [9]. Multiple mechanisms have been reported to be responsible for the deregulation of BCL-2 protein family [9]. The development of strategies to target these pro-survival factors in melanoma has been a central theme for years [10], and may represent an alternative option to defeat melanoma as well as to overcome resistance to current targeted therapies [9].

This scenario supports the rationale for drug combination approaches [2] or, alternatively, for the use of single multi-targeting drug molecules, which are arising as valuable alternative tools to therapeutic regimens based on drug combinations [11], in order to overcome drug resistance and hopefully obtain long-term responses.

Nucleic acids can fold into several structural motifs to assemble the functional structural conformation for their precise biological roles in specific cellular environments. In particular, guanine (G)-rich sequences can self-associate into stacks of G-quartets using Hoogsten-type hydrogen bonds to form complex secondary structures knows as G-quadruplexes (G4s) [12], which are stabilized by K^+^ cations under physiological conditions [13]. In recent years, G4s have attracted great attention, largely due to both their peculiar polymorphisms [14] and critical regulatory roles in biological processes [15], such as modulation of gene expression [16], regulation of epigenetic modifications [17], telomerase dysfunction [18], transcription [19], genomic instability [20], and histone modifications [21]. Their implication in the pathogenesis of cancer [22] and neurodegenerative diseases [23,24] was extensively described, providing new possible targets in a number of different pathologies. In vivo formation of G4s was consolidated by the discovery of cellular proteins that specifically process G4s [25,26] and the development of G4-specific antibodies [27,28] that allowed both direct visualization and chromatin immunoprecipitation (ChIP) sequencing-based approaches [29,30,31]. Besides telomeres [32], G4s are mainly present in gene promoters and in regions that are close to transcription start sites (TSS) [33]. Based on the observation that G4s are located in the core or proximal promoter of genes related to the six hallmarks of cancer [34], a large number of G4 ligands were designed to target oncogene promoters. The hypothesis that G4 induction and stabilization by small molecules could be exploited to repress gene transcription was first demonstrated with the porphyrin ligand TMPyP4, which was able to down-regulate MYC expression [35], paving the way for the quest of specific G4-stabilizing molecules. Since then, many G4 ligands have been developed [36,37], with some showing interesting biological activity and proceeding to clinical trials [38,39]. These agents belong to different chemical families. Among them, naphthalene diimides (NDIs) are a class of small molecules that recognize, induce, and stabilize G4s with high affinity. Thanks to its electron-deficient aromatic systems, the NDI core is prone to interact via π–π stacking with an electron-rich partner, like the G-quartet [40]. Moreover, the extension of the NDI core offers an opportunity to enhance both G4 binding potency and selectivity [41,42]. In the present investigation, we explored the possibility to interfere with oncogenic signaling pathways in BRAF-mutant melanoma cells through NDI-based G4 ligands. A small library of core-extended NDIs (c-exNDIs **1**–**14**, Appendix A) was screened in terms of cytotoxicity on A-375 melanoma cells, and the activity of the most active compound (**1**) was further investigated. Combining biophysical and molecular methods, we proved that **1** greatly stabilized G4s within human oncogene promoters. At the cellular level, the exposure of *BRAF*-mutant melanoma cells to the compound resulted in a marked down-modulation of *KIT*, which was paralleled by the shutdown of the MAPK and PI3K/AKT signaling pathways, both at the transcriptomic and post-translational levels. Moreover, **1**-treated cells were characterized by an expression profile of BCL-2 family members indicating a trend toward a pro-apoptotic environment. We thus present a suitable candidate for the development of novel treatment options based on G4 targeting, which may be particularly relevant in the context of resistance to targeted therapies of *BRAF*-mutant melanoma cells.

## 2. Materials and Methods

### 2.1. Materials and General Procedures

Reagents, solvents, and chemicals were purchased from TCI (Milan, Italy) or Sigma-Aldrich (Milan, Italy) and used as supplied, without further purification. All oligonucleotides (Appendix A) were purchased from Sigma-Aldrich (Milan, Italy). The tested compounds were dissolved in dimethylsulfoxide (DMSO), stored at −20 °C, and diluted in culture medium at the appropriate working concentrations immediately before use. The eluent for all of the HPLC analyses and purifications was 0.1% trifluoroacetic acid in water and acetonitrile. HPLC analysis was performed using Agilent system SERIES 1260. The column was XSelect HSS C18 (2.5 μm) (50 × 4.6 mm) (Waters, Milan, Italy). We used two different methods for analytical and preparative purposes. Analytical method: flow—1.4 mL/min; gradient—95% aqueous for 2 min, gradually to 100% of acetonitrile over 14 min and then isocratic flow for 4 min. Preparative HPLC purifications were attained using Agilent system SERIES 1260 and equipped with a preparative PUMP, a diode array system, and an automatic fraction collector. The preparative column was XSelect CSH Prep Phenyl-Hexyl 5 μm (150 × 30 mm) (Waters, Milan, Italy) using a 30 mL/min flow. The eluent was water + 0.1% trifluoroacetic acid, gradually decreased from 95 to 85%, over 4 min, then gradually to 65% water in 16 min. ^1^H- and ^13^C-NMR spectra were recorded on a Bruker ADVANCE 300 MHz.

### 2.2. Cell Lines

Human tumor cell lines and primary dermal fibroblasts from adult skin were obtained as indicated in Appendix A. Cells were maintained as a monolayer in the logarithmic growth phase in the appropriate growth medium in the presence of 10% fetal calf serum at 37 °C in a humidified incubator at 5% CO_2_. Cells were periodically monitored for DNA profiling by short tandem repeats analysis using the AmpFISTR Identifiler PCR amplification kit (Thermo Fisher Scientific, Monza, Italy).

### 2.3. Cell Viability Assay

The MTT (3-(4,5-dimethylthiazol 2-yl)-2,5-diphenyltetrazolium bromide, Sigma-Aldrich, Milan, Italy) assay was performed to assess the cytotoxicity of c-exNDIs on a human malignant melanoma A375 cell line. A375 cells (2.5 × 10^3^) were plated in 96-well plates and incubated for 24 h. Cells were treated with increasing concentrations (1.95–500.00 nM) of tested compounds and incubated for an additional 48 h. Cell viability was evaluated by MTT assay by adding 10 μL of freshly dissolved solution of 5 mg/mL MTT in phosphate buffered saline (PBS) to each well. After 4 h of incubation, MTT crystals were solubilized in solubilization solution (10% sodium dodecyl sulfate (SDS) and 0.01 M HCl). After overnight incubation at 37 °C, absorbance was read at 570 nm. The percentage of cell viability was calculated as follows: Cell survival = (A_well_ − A_blank_)/(A_control_ − A_blank_) × 100, where blank denotes the medium without cells. The IC_50_ was defined as the concentration of compound required to inhibit cell grown by 50%. Data were expressed as mean values ± standard deviation of three independent experiments.

### 2.4. Gene Expression Analysis

Total RNA was isolated using Qiagen RNeasy Mini kit (Qiagen S.r.l., Milan, Italy) and digested with 20 U RNase-free DNase I (Qiagen S.r.l., Milan, Italy), according to the manufacturer’s instructions. Five hundred nanograms of total RNA was randomly primed and reverse transcribed using the High Capacity cDNA Reverse Transcription kit (Applied Biosystems, Carlsbad, CA, USA). The expression levels of 92 genes associated with the molecular mechanisms of cancer were assessed by real-time RT-PCR using TaqMan Arrays (PN4418806; Applied Biosystems), according to the manufacturer’s instructions. Amplifications were run on 7900HT fast real-time PCR system. Data were analyzed by SDS 2.2.2 software and, if not otherwise specified, reported as relative quantity (RQ) with respect to untreated cells (calibrator sample), according to the 2^−ΔΔCt^ method [43], where Ct represents the threshold cycle. The 18S housekeeping gene present in each array was used as a normalizer. Differentially expressed genes in treated vs. untreated cells were sorted based on *p* < 0.05 (Student’s t–test) and considered up- or down-regulated by setting a fold-change of 1.5 as the cut-off.

### 2.5. Circular Dichroism

For circular dichroism (CD) analysis, oligonucleotides (Appendix A) were diluted to a final concentration (4 μM) in lithium cacodylate buffer (10 mM, pH 7.4) and KCl 10 mM. Samples were annealed by heating at 95 °C for 5 min and gradually cooled to room temperature and, where indicated, **1** was added at the final concentration of 16 μM. CD spectra were recorded using Chirascan-Plus (Applied Photophysisics, Leatherhead, United Kingdom) equipped with a Peltier temperature controller using a quartz cell of 5 mm optical path length, over a wavelength range of 230–320 nm. The reported spectrum of each sample represented the average of 2 scans at 20 °C and was baseline corrected for signal contributions due to the buffer. Observed ellipticities were converted to mean residue ellipticity (θ) = deg × cm^2^ × dmol^−1^ (molar ellipicity). For the determination of melting temperature (T_m_), spectra were recorded over a temperature range of 20–90 °C, with temperature increase of 5 °C. T_m_ values were calculated according to Greenfield [44].

### 2.6. Fluorescence Resonance Energy Transfer

For fluorescence resonance energy transfer (FRET), 6-carboxyfluorescein (FAM) 5′-end-labeled and 6-carboxy-tetramethylrhodamine (TAMRA) 3′-end-labeled oligonucleotides (0.25 μM) (Appendix A) were folded in lithium cacodylate buffer (10 mM, pH 7.4) and KCl 100 mM or 10 mM by heating at 95 °C for 5 min and gradually cooling to room temperature. Where indicated, **1** was added at a final concentration of 0.25 μM and, after stabilization at 4 °C, samples were processed in a Light Cycler (Roche, Milan, Italy). Oligonucleotide melting was monitored by measuring FAM emission in the 30–95 °C temperature range with a gradient of 1 °C/min. Melting profiles were normalized as previously described [45] and T_m_ was defined as the temperature corresponding to the 0.5 fraction of the normalized fluorescence.

### 2.7. Taq Polymerase Stop Assay

The DNA primer (Appendix A) was 5′-end labeled with [γ-^32^P]ATP using T4 polynucleotide kinase (Thermo Scientific, Milan, Italy) at 37 °C for 30 min. The labeled primer (final concentration 72 nM) was annealed to the templates (final concentration 36 nM) (Appendix A) in lithium cacodylate buffer (10 mM, pH 7.4) in the presence or absence of KCl 10 mM by heating at 95 °C for 5 min and gradually cooling to room temperature to allow both primer annealing and G4 folding. Where specified, samples were incubated with **1** (37.5–150.0 nM) and primer extension was performed with 2 U/reaction of AmpliTaq Gold DNA polymerase (Applied Biosystem, Carlsbad, CA, USA) at 56 °C for 30 min. Reactions were stopped by ethanol precipitation and primer extension products were separated on a 14% denaturing gel, and visualized with phosphorimaging (Typhoon FLA 9000, GE Healthcare, Milan, Italy). Markers were prepared based on Maxam and Gilbert sequencing by PCR reaction with ^32^P-labeled primer. PCR products were treated with formic acid for 5 min at 25 °C and then with piperidin for 30 min at 90 °C.

### 2.8. Western Immunoblotting

Total protein extracts were prepared according to standard methods. Forty micrograms of protein extract was fractioned by SDS-PAGE and transferred onto Hybond nitrocellulose filters (RPN 303D, GE Healthcare, Milan, Italy). Filters were blocked in PBS-Tween 20 and 5% skim milk and incubated overnight with the following primary antibodies: mouse monoclonal anti-BCL-2 (sc-509, Santa Cruz Biotechnology, Dallas, TX, USA), anti-KIT (#3308; Cell Signaling Technology, Danvers, MT, USA), and anti-MYC (ab32; Abcam, Cambridge, United Kingdom); rabbit polyclonal anti-AKT(S473) (sc-9271, Santa Cruz Biotechnology), anti-BRAF (ab33899, Abcam,), anti-ERK1/2 (T202/Y204) (#9101, Cell Signaling Technology), anti-γ-H2AX (ab11174, Abcam), anti-p21^waf1^ (ab7960, Abcam), and anti PARP-1 (#9542, Cell Signaling Technology). Mouse monoclonal anti-Vinculin (VCL, V9131, Sigma-Aldrich, Milan, Italy) or anti-β-Actin (Ab8226, Abcam) antibodies were used to ensure equal protein loading. The filters were then probed with secondary peroxidase-linked whole antibodies (GE Healthcare, Milan, Italy) and subjected to autoradiography using the Novex^®^ Enhanced Chemoluminescent Horseradish Peroxidase detection system (Thermo Fisher Scientific, Monza, Italy). Films were scanned (ImageScanner III, GE Healthcare, Milan, Italy) and images were processed by Photoshop7.0.1 or analyzed using ImageJ 1.46r.

### 2.9. Analysis of Cell Cycle Phase Distribution

For the cell cycle phase distribution analysis, both adherent and floating cells were fixed in 70% EtOH and incubated at 4 °C for 30 min in staining solution containing 50 mg/mL of propidium iodide, 50 mg/mL of RNase, and 0.05% Nonidet-P40 in PBS. Samples were analyzed with a Fluorescence Activated Cell Sorting (FACS)-Calibur cytofluorimeter (Becton Dickinson, Milan, Italy). At least 30,000 events were read, and the histograms were analyzed using the CellQuest software according to the Modfit model (Becton Dickinson, Milan, Italy).

### 2.10. Statistical Analysis

If not otherwise specified, the data were reported as mean values ± s.d. from at least three independent experiments. Two-sided Student’s *t*-test was used to analyze any differences between samples. The association between the IC_50_ values and the cell doubling times in the cancer cell lines was assessed using the Spearman rank order correlation coefficient (GraphPad Software Inc., San Diego, CA, USA). *p* values of <0.05 were considered statistically significant.

## 3. Results

### 3.1. Chemistry

A family of fourteen c-exNDIs (Appendix A) characterized by a lateral extension of the NDI core was selected to be screened against melanoma cells. The compounds were designed to expand the π-stacking interaction with the G-quartet moiety in comparison to NDIs, thereby investigating whether binding and affinity to G4s were improved. Compounds **1**–**10**, **13**, and **14** were previously synthesized and characterized [41,42], while **11** and **12** were novel ligands and were prepared according to the synthetic protocol outlined in Appendix A. The screening and biophysical assays depicted below were performed using the hydrochloride amine salts (**1**–**7** × 2 HCl, and **11**–**14** × 2 HCl) and the iodide quaternary ammonium salts **9** and **10**.

### 3.2. Screening of the Anticancer Activity of c-exNDIs in Melanoma Cells

We first assessed the cytotoxic activity of the c-exNDI series of ligands via MTT assay in a representative in vitro model of mutant *BRAFV600E* melanoma. Specifically, 48 h of exposure of A375 cells to increasing concentrations of each compound resulted in a remarkable and dose-dependent inhibition of cell growth, with IC_50_ values in the nanomolar range for most compounds (Table 1). Among the tested c-exNDIs, the unsubstituted ligands at C4 (Y = H, see Appendix A for numbering) were generally more active compared to the bromine derivatives (Y = Br). **1**, **2, 3,** and **11** displayed the highest cytotoxicity levels, with IC_50_ values in the low nM range. On the other hand, **9** and **10**, which contained two quaternary ammonium moieties, did not display any cytotoxic effects up to the highest tested concentration, likely due to their inability to enter the cells, as already shown for NDIs with permanent positive charges [46]. Low activity was also detected with **5** and **6**, which exhibited IC_50_ values higher than 500 nM, probably due to the anionic carboxylate moiety. In general, substituents at C11 had detrimental effects on cytotoxicity. The other c-exNDIs showed intermediate values of cytotoxicity.

Compound **1**, which was the most active one in the series (IC_50_ of 8 ± 1 nM, Table 1), was selected and assessed against a panel of various cell lines representative of tumors of different histological origins. As reported in Appendix A, the compound exhibited a pronounced cytotoxic activity on the tested tumor cell lines, whereas it did not remarkably cause cytotoxic effects on normal human primary skin fibroblasts at concentrations within the nanomolar range.

This evidence was in keeping with our previous findings showing that the compound exhibited a pronounced cytotoxic activity in a panel of metastatic castration-resistant prostate cancer cells, while sparing normal prostate epithelial cells [47] and thus contributing to the delineation of a therapeutic window for the compound.

### 3.3. C-exNDI 1 Induces a Global Down-Regulation of Gene Expression in BRAFV600E-Mutant Melanoma Cells

It was hypothesized that the effects at the transcriptomic level of small molecule-mediated stabilization of G4 located within gene transcriptional regulatory elements could be more pronounced in rapidly dividing cells than in slowly dividing cells. In this context, Spearman rank correlation analysis revealed the occurrence of a possible relationship between the cytotoxic activity of compound 1 and the doubling times for the 9 cancer cell lines listed in Appendix A. Specifically, a statistically significant trend toward a strong, positive monotonic correlation between IC_50_ values and the cell doubling times was revealed (r_s_ = 0.8201, *n* = 9, *p* = 0.0108). This observation indicated that G4 formation most likely occurs during DNA replication [15], as shown by the evidence that the number of G4 structures increased in human cells during the S-phase of the cell cycle when the duplex strands became separated at the replication forks, potentially causing single-stranded DNA to fold more easily into secondary structures [28], whereas it decreased following the exposure to aphidicolin, an inhibitor of DNA replication [28].

A comparative analysis of gene expression was carried out on a pair of malignant melanoma cell lines (A375 and SKMEL-2), which was characterized by both a distinct rate of growth in vitro and a pronouncedly different sensitivity to **1** (Figure 1A,B). Results showed that 48 h of exposure to an equitoxic amount (IC_50_) of the compound triggered G4 stabilization (Figure 1C) and induced a global down-regulation of gene expression in A375 compared to SKMEL-2 cells (Appendix A). In particular, real-time RT-PCR assessment of the expression levels of 92 genes commonly associated with the molecular mechanisms of cancer revealed that ~61% of genes were differently expressed (fold-change > |1.5|) in **1**-treated compared to untreated A375 cells (e.g., 53 and 3 genes were down- and up-modulated, respectively, Appendix A). Conversely, only ~22% of genes were found to be differently expressed (10 down- and 10 up-regulated) in **1-**treated SKMEL-2 (Appendix A). Notably, an over-representation enrichment analysis carried out using the GEne SeT AnaLysis Toolkit (WebGestalt, http://webgestalt.org) revealed that the most relevant over-represented pathways (*p* < 0.01) defined by the list of differently expressed genes in **1-**treated vs. untreated A375 cells with a fold-change >|2.0| comprised terms such as signaling by BRAF and RAF fusions, oncogenic MAPK signaling, and signal attenuation (Appendix A), which are all relevant, targetable pathways in *BRAF*-mutant melanoma cells [3]. In more detail, real-time RT-PCR data showed a marked decrease in the expression levels of factors associated with RAS/RAF/MAPK and PI3K/AKT signaling pathways, as well as significant modulations in the expression levels of BCL-2 family members, indicating a trend toward a pro-apoptotic milieu, with down-regulation of the anti-apoptotic *BCL2* and *BCL2L1* transcripts and the up-regulation of pro-apoptotic factors, such as *BAX* and *BCL2L11* (Figure 1D). On the contrary, although the paucity of differently expressed genes showing a fold-change >|2.0| did not allow us to retrieve significantly over-represented pathways (Appendix A), it was clear that the exposure of SKMEL-2 cells to **1** resulted in a completely different gene expression landscape, without any remarkable modulation in the expression levels of factors belonging to RAS/RAF/MAPK except for the down-modulation of *MAP3K5* (0.65 ± 0.03; Appendix A) and/or PI3K/AKT pathways, and no significant perturbations in the levels of BCL-2 family members, except for a marked down-modulation of the *BID* transcript (0.47 ± 0.005; Appendix A).

The different pattern of gene expression modulation observed in melanoma cell lines exposed to compound **1** was paralleled by a distinct response in terms of DNA damage induction. In particular, similarly to cells exposed to pyridostatin, which was used as a reference G4 ligand, the exposure of A375 cells to **1** resulted in a marked induction of DNA damage, as suggested by the prominent occurrence of DNA damage foci observed by immunofluorescence in cells probed with anti-γ-H2AX antibody (Appendix A). On the other hand, and differently from the pyridostatin treatment, **1**-treated SKMEL-2 cells did not show remarkable accumulation of γ-H2AX foci (Appendix A).

Notably, it could not be excluded that the different observed scenarios between A375 and SKMEL-2 cells in terms of **1**-dependent gene expression modulation represented a distinct cell response to the compound that reflected their opposite genetic background (*TP53^WT^*, BRAF^V600E^, and *NRAS^WT^* in A375 cells, and *TP53*^G245S^, *BRAF^WT^*, and *NRAS^Q61R^* in SKMEL-2 cells; see COSMIC, the Catalogue Of Somatic Mutations In Cancer, https://cancer.sanger.ac.uk/cell_lines). Nonetheless, it is worth nothing that exposure of *BRAF^WT^/NRAS^MUT^* cells to **1** resulted in a peculiar gene expression pattern with over-represented pathways, including platelet degranulation, senescence-associated secretory phenotypes, and oxidative stress-induced senescence (Appendix A). This scenario suggested that **1-**treated SKMEL-2 cells underwent changes at the cellular/molecular level, likely resembling phenotype switching [1]. This phenomenon of cell plasticity, which includes, among others, epithelial-to-mesenchymal-like transition, switching from differentiated to de-differentiated, and slowing proliferating phenotypes, as well as metabolic rewiring (including increased Reactive Oxygen Species production), represented an adaptive process in response to treatment-induced insults by which melanoma cells may adapt to therapies [1].

### 3.4. Melanoma-Relevant Genes as Possible G4 Targets of c-exNDI 1

The finding that the exposure of *BRAF*-mutant melanoma cells to our G4 ligand resulted in modulation of the expression levels of genes belonging to signaling pathways that are relevant to melanoma cell survival and drug-resistance prompted us to investigate whether the compound interacted with specific G4 targets at the interplay of RAS/RAF/MAPK and PI3K/AKT, as well as the intrinsic apoptosis pathways. In particular, we focused on *KIT*, *BRAF,* and *BCL-2*, three relevant target genes in melanoma known to bear G4-forming sequences within their promoters [48]. Using biophysical and biomolecular assays, we comparatively assessed the ability of **1** to stabilize the G4-forming sequences within the promoter regions of the selected genes in comparison to *MYC*, which is known for having the highest G4 density among oncogenes [40].

Five different oligonucleotides corresponding to the G4-forming sequences within the promoter regions of *MYC*, *BRAF*, *BCL-2,* and *KIT* (*KIT-1* and *KIT-2*) (Appendix A) were assayed by circular dichroism (CD) to analyze both their conformation and their stability in the presence and absence of the compound. In-keeping with data reported in the literature and concerning the biophysical characterization of G4 motifs of the selected genes [49,50,51,52,53], we found that the CD spectra of the tested oligonucleotides were all distinctive of G4 topologies. In particular, the c-myc and c-kit2 sequences formed a parallel-type G4 structure showing a major positive peak at around 264 nm and a negative one at 242 nm. A prevalent parallel conformation was also observed in the bcl-2, c-kit1, and b-raf templates, the last two displaying an additional positive shoulder at around 290 nm, indicating a mixed or hybrid G4 conformation (Figure 2A). Upon addition of **1**, c-myc, bcl-2, c-kit1, and c-kit2 maintained a prevalent parallel conformation, with the molar ellipticity at 264 nm of c-myc, c-kit1, and c-kit2 decreasing to different extents (Figure 2B). Additionally, a slight variation in the molar ellipticity signal was observed in bcl-2 and c-kit1 at 290 nm (an increase and a decrease, respectively). In contrast, **1** induced a structural transition from a parallel-type to a mixed parallel–antiparallel G4 on the b-raf template, as indicated by the presence of two positive peaks at 264 and 290 nm (Figure 2B).

Stability of the G4 structures was next evaluated by CD thermal unfolding (Appendix A); the melting temperatures (T_m_) are reported in Table 2. For c-myc, bcl-2, c-kit1, and c-kit2, a single transition between 20 °C and 90 °C was appreciable at 264 nm, leading to discrete T_m_ values. A remarkable increase in stability with significant variation of the T_m_ (>90 °C) was recorded in the presence of the compound compared to the absence of the compound. Given that **1** induced a mixed G4 on the b-raf templates, T_m_ was calculated at both 264 nm and 290 nm; a notable increase in the T_m_ was recorded (Table 2) upon addition of **1**, indicating effective stabilization of b-raf G4.

We next employed a dual-labeled system. The stabilizing activity of **1** was assessed by Fluorescence Resonance Energy Transfer (FRET) melting assay on c-kit2 (Appendix A), which was chosen as the representative oncogene G4-forming sequence. C-kit2 displayed a T_m_ of 76.4 ± 0.3 °C and 65.7 ± 0.7 °C at 100 mM and 10 mM of K^+^, respectively. In both conditions, c-kit2 was greatly stabilized by **1** with T_m_ > 90 °C (Appendix AA,B). A double stranded (ds) oligonucleotide (Appendix A) was also used as a non-G4-forming sequence control to check the selectivity of **1** for G4 vs. the duplex. This showed a T_m_ of 65.7 ± 0.7 °C and 61.2 ± 01 °C at 100 mM and 10 mM of K^+^, respectively, with the compound showing that it was basically devoid of a stabilizing effect on dsDNA with variations in T_m_ values of <2 °C (Appendix AC,D).

To corroborate the ability of **1** to stabilize the target G4s, a Taq polymerase stop assay was performed using bcl-2, c-myc, b-raf, and c-kit1+2 templates, the latter containing both c-kit1 and c-kit2 G4-forming motifs linked by a 16 nucleotide-long sequence (Appendix A). All tested oligonucleotides were designed to contain additional flanking bases at both the 5′- and 3′-ends; in particular, an additional sequence at the 3′-end was used as a primer annealing region and 5-T linker regions were added to both the 3′-end, to separate the primer annealing region and the first G of the G4 portion, and the 5′-end. All G4-forming sequences, except for b-raf, stopped the polymerase at the first G-rich region encountered by the polymerase, even in the absence of K^+^ (Figure 2C, lane 1: bcl-2, c-myc, c-kit1+2). In the presence of K^+^ (Figure 2C, lane 2: b-raf, bcl-2, c-myc, c-kit1+2), stop sites corresponding to the first 3′G-tract were visible in all templates and were more intense than in the absence of K^+^, thus indicating that K^+^ stimulated G4 folding. Upon addition of increasing amounts (37.5–150.0 nM) of **1**, the intensity of the stop bands greatly rose in all templates in a concentration-dependent manner (Figure 2C, lanes 3–5), accompanied by a considerable reduction in full-length amplicons, thus corroborating the effective stabilization of the G4s by the compound at nM concentrations. Furthermore, additional stop sites corresponding to other G-rich regions were observed in bcl-2, c-myc, and c-kit1+2 templates, possibly indicating the presence of multiple G4 structures. Quantification of the stop sites confirmed the concentration dependence of polymerase stalling of the compound (Figure 2D). No stop sites were observed with a control sequence devoid of G-tracts (Figure 2C, lanes 1–4: non-G4 cnt), indicating that the observed polymerase inhibition was G4-dependent. Overall, these data were in line with those obtained by CD analysis and confirmed the ability of the chosen sequences to be stabilized by **1**.

### 3.5. The Exposure of BRAFV600E Melanoma Cells to c-exNDI 1 Results in KIT and BCL-2 Protein Down-Regulation, Inhibition of MAPK and PI3K/AKT Signaling Pathways, and Apoptosis Induction

Gene expression analyses revealed a reduction in the expression levels of *BRAF, BCL-2,* and *MYC* mRNAs in A375 cells upon 48 h of exposure to the ligand (Figure 1D and Figure 3A), whereas the levels of *KIT* mRNA remained undetermined both in treated and untreated cells, being undetectable even after 40 cycles of PCR amplification (Appendix A). Nonetheless, the validation of qRT-PCR data by Western immunoblotting revealed a time-dependent reduction of both KIT and Bcl-2 protein amounts in **1-**treated A375 cells with respect to untreated A375 cells (Figure 3B), whereas MYC and BRAF protein levels were not remarkably affected by the treatment (Figure 3B). On the other hand, *KIT* and *MYC* mRNA levels were mildly reduced (RQ = 0.67 ± 0.05 and 0.74 ± 0.03, respectively) in SKMEL-2 cells exposed for 48 h to an equitoxic amount of the compound (Figure 3A and Appendix A).

The observed reduction in *KIT* protein levels was accompanied by perturbations of downstream signaling pathways. In particular, 72 h of exposure of A375 cells to the compound resulted in a mild reduction (*p* < 0.05) of phosphorylated MAPK3 (p44/ERK1^T202^)—although with no modification in the phosphorylation status of MAPK1 (p42/ERK2^Y204^)—as well as in a pronounced decrease (*p* < 0.01) in the phosphorylation of AKTS473 (Figure 3C). As already observed, **1**-treated cells were characterized by the prominent accumulation of γ-H2AX (Figure 3C) and by a marked increase in the amounts of the cyclin-dependent kinase inhibitor p21^waf1^ (Figure 3C). Consistent with gene expression data (Appendix A), a moderate increase in p21^waf1^ protein levels was also appreciable in SKMEL-2 cells exposed to an equitoxic amount of the compound (Appendix AB). Conversely, **1**-treated SKMEL-2 cells were characterized by an increase in the phosphorylated forms of ERK1/2 and AKT, without showing a remarkable induction of DNA damage (Appendix A).

In addition, exposure to **1** led to significant perturbations in the progression of A375 cells throughout the cell cycle, as suggested by the significant increase over time in the percentage of cells residing in the G1 phase in treated vs. untreated cells (Figure 3D). Finally, in keeping with BCL-2 protein reduction (Figure 3B) and a trend in the expression of BCL-2 family transcripts suggesting a pro-apoptotic framework (Figure 1D), the exposure of A375 cells to **1** resulted in apoptosis induction, as revealed by the significant increase in the amounts of the cleaved form of PARP1 after 48 and 72 h of drug exposure compared to untreated cells (Figure 3E).

## 4. Discussion

Melanoma comprises a spectrum of malignancies characterized by targetable mutations in distinct oncogenic signaling pathways [3,9]. The development of BRAF and MEK inhibitors as treatment options for patients with *BRAF*-mutant tumors has contributed to improve the median overall survival of patients with metastatic melanoma [1,3]. Nevertheless, although these treatments lead to a quick initial response, most patients eventually relapse due to primary or secondary resistance mechanisms [1]. To date, distinct resistance mechanisms to targeted therapies have been described. These comprise the reactivation of MAPK signaling through *NRAS* mutations or amplification or alternative splicing of *BRAF* and *MEK1/2* mutations [8]. Moreover, resistance may occur in a MAPK-independent fashion via the activation of alternative pathways (e.g., PI3K/AKT), the over-activation of tyrosine kinase receptors, and the evasion of apoptosis [1,8,9]. Hence, despite the advent of targeted therapies, melanoma still remains incurable and represents a tumor that is very difficult to treat [1,2].

In the last decades, G4 structures have emerged as attractive candidates for cancer therapy [22,48]. The growing interest in these oncogene promoters as therapeutic targets has fueled the rational design and development of small molecules that are able to recognize and stabilize such secondary structures [54,55]. A variety of G4-stabilizing small molecules identified in recent years are currently considered as fascinating weapons to therapeutically operate at the genomic level and represent a novel, though still challenging, tool to target oncogene promoters and downstream events in cancer cells.

In the present study, we showed that **1** was able to bind and stabilize G4 structures in vitro that form within the promoters of human *KIT*, *BRAF,* and *BCL-2*, which are three genes that may be relevant therapeutic targets in melanoma. Specifically, the biophysical data obtained by CD indicated that c-kit1, c-kit2, b-raf, and bcl-2 G4s were greatly stabilized by the compound to similar extents, with T_m_ values of >90 °C. Moreover, the Taq polymerase stop assay corroborated the ability of these sequences to be stabilized by **1** and also stalled polymerase progression.

At the cellular level, we observed that the exposure of *BRAF*-mutant melanoma cells to **1** resulted in a pronounced down-modulation of *KIT*, which was paralleled by the shutdown of the RAS/RAF/MAPK and PI3K/AKT signaling pathways, both at the transcriptomic and post-translational levels. Moreover, **1**-treated cells were characterized by an expression profile of BCL-2 family members indicating a trend toward a pro-apoptotic milieu, which was also confirmed at the biochemical level by the remarkable reduction of BCL-2 protein amounts and the prominent accumulation of γ-H2AX and p21^waf1^, as well as the cleaved form of PARP-1. Conversely, our data revealed that *NRAS*-mutant melanoma cells were not particularly affected in oncogene-mediated signaling pathways, but rather showed a gene expression pattern in response to **1** exposure that was likely consistent with a phenotypic switch [1].

Altogether, our results supported the notion that the transcriptional activation of oncogenes in *BRAF*-mutant melanoma cells could be suppressed through the stabilization of G4s within gene promoters, and indicated that G4-stabilizing compounds could be used to interfere with oncogene expression, especially KIT and BCL-2, in a “double-hit” manner. However, we could not exclude that additional direct targets may have been amenable to **1**-mediated G4 stabilization and, in keeping with previous evidence [56], our data suggested that the cell response to G4 ligands was largely dependent on the cellular context, as well as on the presence of specific genetic alterations.

Nonetheless, given the multifactoriality of cancer and the involvement of G4s in all its hallmarks [34], the low grade of specificity of the ligand toward one single G4 may be viewed as an advantage [22], and could be exploited to suppress the proliferation of tumor cells through activity at multiple targets. In fact, as a large body of preclinical and clinical evidence demonstrates, highly aggressive tumors such as melanomas may not be able to be defeated by monotherapies; rather, combinations of several signaling effectors should be envisioned [57].

While KIT may not represent a reliable primary therapeutic target in melanoma due to it being mutated in a small fraction of lesions [1,3], its relevance as a target may be envisaged in the context of acquired drug resistance of *BRAF*-mutant melanoma treated with targeted therapies. In fact, KIT activates a series of downstream effectors, such as the MAPK pathway with the activation of MAPK3/1 (ERK1/2) and the PI3K/AKT pathway [7]. Of note, the aberrant activation of these pathways plays a major role in conferring resistance to cell death (i.e., drug-induced apoptosis) in melanoma [7,9]. Notably, pro-death proteins of the BCL-2 family, such as BAD, BAX, BID, and BCL2L11, are inhibited by both AKT and MEK, whereas pro-survival factors belonging to the same family, such as BCL2L1, MCL1, and BCL-2, are activated by the MAPK pathway [7,9]. In this context, activation of MAPK or PI3K/AKT signaling pathways contributes to the relapse of melanoma patients treated with BRAF inhibitors [9]. Moreover, it was reported that BRAFV600E suppressed apoptosis by up-regulating the anti-apoptotic factor MCL1, whereas constitutively activated RAS/BRAF/MAPK signals inactivated the pro-apoptotic proteins BAD and BCL2L11. In addition, the activated PI3K/AKT pathway inhibited cell death by preventing the binding of BAD to BCL-2 or BCL2L1 and impeded the induction of apoptosis upon BRAF inhibition. Consequently, taking into account that BCL-2 family members may undergo regulation by MAPK or PI3K/AKT signaling pathways and that the activation of these pathways plays a key role in BRAF-inhibitor resistance, targeting downstream apoptotic proteins may represent a suitable option to overcome relapses [9]. In this context, different strategies to target Bcl-2 family members with promising pre-clinical evidence were reported [9]. For instance, it was shown that inhibiting BCL-2 in combination with MCL-1 inhibition or NOXA activation represented an effective approach to induce melanoma cell death, as well as that targeting BCL-2 with small molecule inhibitors (e.g., the BH3-mimetic ABT-737) sensitizes melanoma cells to BRAF inhibitors [9].

## 5. Conclusions

Overall, by affecting the expression levels of both KIT and BCL-2 proteins and, consequently, by impinging on MAPK and PI3K/AKT pathways, as well as on the balance between pro-death and pro-survival factors, our compound may be used to interfere with oncogene expression in melanoma cells. We thus present a starting point for the future development of therapeutic interventions based on G4 targeting that may be of particular relevance in the context of resistance to targeted therapies of *BRAF*-mutant melanoma cells.

## Figures and Tables

**Figure 1 cells-08-01274-f001:**
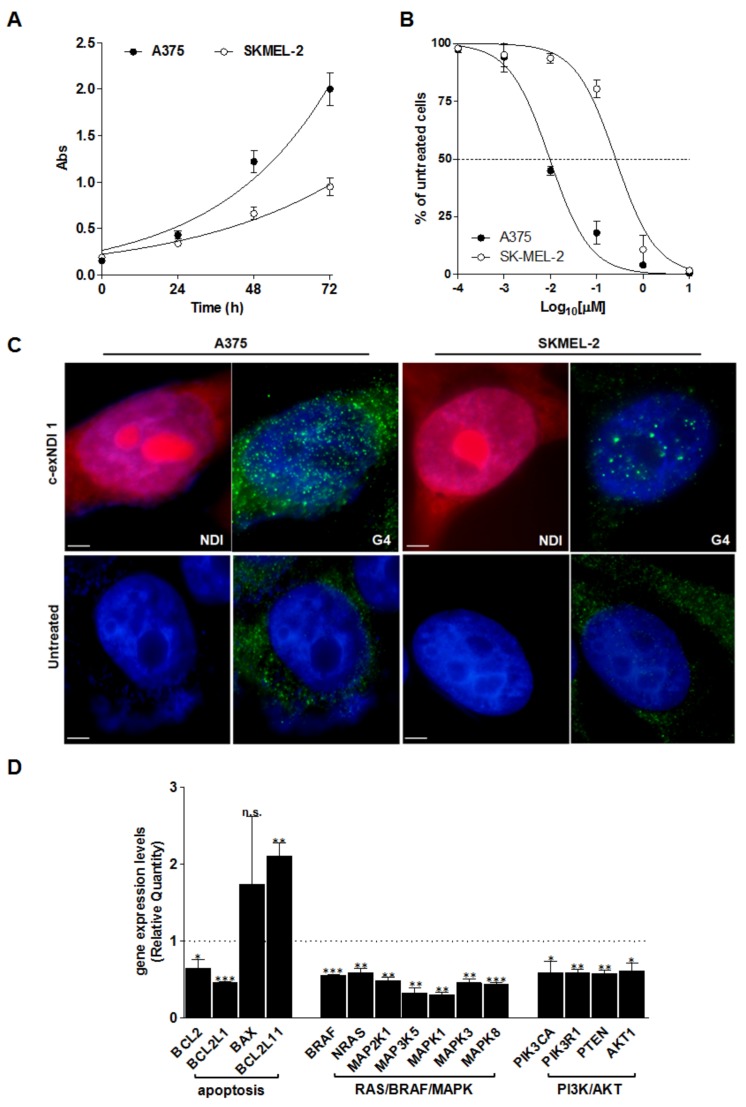
(**A**) Time-dependent assessment of cell growth in untreated A375 and SKMEL-2 cells. Data are reported as the recorded absorbance (Abs) as a function of time and represent mean values ± s.d. from at least three independent experiments. The curves represent the exponential growth equation obtained by nonlinear fit of data points using GraphPad Prism 5.01 (GraphPad Software Inc., San Diego, California, United States). (**B**) Dose-response curves of melanoma cells exposed to increasing concentrations (0.1–10,000 nM) of **1**. Data are reported as the percentage of viable cells with respect to untreated cells as a function of the Log_10_ of compound concentrations (for details see Appendix A) and represent mean values ± s.d. from at least three independent experiments. (**C**) Representative photomicrographs showing the uptake of the G4 ligand (red) and the occurrence of G4 structures (green) in A375 and SKMEL-2 cells exposed to c-exNDI 1 (IC_50_) for 48 h and analyzed by fluorescence microscopy. The panels on the bottom show untreated cells. Nuclei were counterstained with DAPI. Magnification: ×100; bars: 10 μm. (**D**) Analysis of gene expression levels in A375 cells upon 48 h of exposure to **1** (IC_50_). Data are reported as Relative Quantity (mean RQ values ± s.d.) in treated vs. untreated cells, according to the 2^−ΔΔCt^ method. Dotted line: calibrator sample. * *p* < 0.05; ** *p* < 0.01; *** *p* < 0.001 (Student’s *t*-test).

**Figure 2 cells-08-01274-f002:**
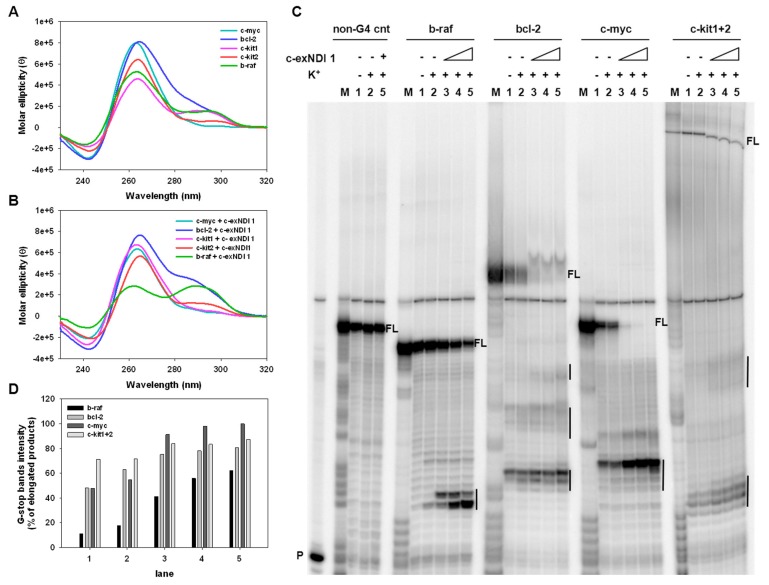
Representative circular dichroism (CD) spectra of G4s formed by c-myc, bcl-2, c-kit1, c-kit2, and b-raf oligonucleotides (4 μM) recorded in the absence (**A**) or presence (**B**) of a 4-fold excess of **1** (16 μM). (**C**) Image of a typical Taq polymerase stop assay. The b-raf, bcl-2, c-myc, and c-kit1+2 templates were amplified by Taq polymerase in the absence (lanes 1) and presence of 10 mM K^+^, alone (lanes 2) or combined, with increasing amounts (37.5, 75.0, and 150.0 nM) of **1** (lanes 3–5). A template (non-G4 cnt) made of a scramble sequence unable to fold into G4 was also used as an internal control. Lane M: Ladder of markers obtained by the Maxam and Gilbert sequencing carried out on the amplified strand complementary to the template strand. Lane P: Unreacted labelled primer. Vertical bars indicate G4-specific Taq polymerase stop sites. (**D**) Quantification of the intensity of G4 stop bands obtained in the Taq polymerase stop assay. G4 stop band intensities are expressed as percentages with respect to the total elongated products.

**Figure 3 cells-08-01274-f003:**
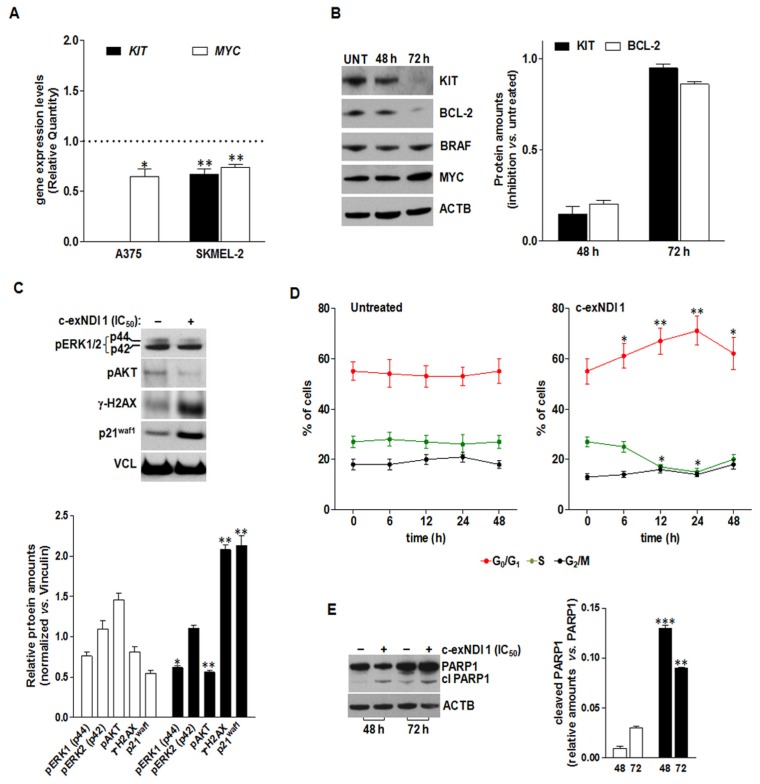
(**A**) Analysis of *KIT* and *MYC* mRNA expression levels in A375 and SKMEL-2 cells upon 48 h of exposure to **1** (IC_50_). Data are reported as Relative Quantity (mean RQ values ± s.d.) in treated vs. untreated cells, according to the 2^−ΔΔCt^ method. Dotted line: calibrator sample. (**B**) Representative Western immunoblotting showing the effects of 48 and 72 h of exposure of A375 cells to **1** (IC_50_) on the amounts of the indicated proteins. Cropped images of selected proteins are shown. Beta-actin (ACTB) was used to ensure equal protein loading. The graph on the right shows the extent of KIT (black bars) and BCL2 (white bars) protein inhibition in A375 cells exposed to **1** (IC_50_) for 48 and 72 h. Data are reported as the inhibition of protein expression in treated vs. untreated cells, and represent the mean values ± s.d. from at least three independent experiments. (**C**) Representative Western immunoblotting showing the amounts of the indicated proteins in untreated (−) A375 cells and upon 72 h of exposure (+) to **1** (IC_50_). Vinculin (VCL) was used to ensure equal protein loading. Cropped images of selected proteins are shown. The graph on the bottom reports the quantification of protein levels in untreated (white bars) and **1**-treated (black bars) A375 cells. Data are reported as relative protein amounts following the normalization of densitometric signals toward Vinculin, and represent the mean values ± s.d. (**D**) Time-dependent assessment of the cell cycle in untreated and **1**-treated A375 cells. Data are reported as the percentages of cells in G0/G1 (red), S (green), and G2/M (black) phases of the cell cycle, and represent the mean values ± s.d. from at least three independent experiments. (**E**) Representative Western immunoblotting showing the amounts of full length and cleaved PARP1 in untreated (−) A375 cells and upon 48 and 72 h of exposure (+) to **1** (IC_50_). Beta-actin (ACTB) was used to ensure equal protein loading. Cropped images of selected proteins are shown. The graph on the right reports the amounts of cleaved PARP1 in untreated (white bars) and **1**-treated (black bars) cells. Data are reported as relative protein amounts vs. full length PARP1, and represent the mean values ± s.d. * *p* < 0.05; ** *p* < 0.01; *** *p* < 0.001 (Student’s *t*-test).

**Table 1 cells-08-01274-t001:** Cytotoxic activity of core-extended naphthalene diimides (c-exNDIs) in A375 melanoma cells.

Compound	IC_50_ ^1^ (nM)
**1**	8 ± 1
**2**	20 ± 2
**3**	50 ± 3
**4**	74 ± 6
**5**	>500
**6**	>500
**7**	101 ± 11
**8**	94 ± 6
**9**	>500
**10**	>500
**11**	54 ± 2
**12**	104 ± 3
**13**	415 ± 36
**14**	195 ± 19

^1^ IC_50_ values (concentration of the compound that inhibited 50% of cell growth) were calculated from the dose-response curves obtained upon 48 h of exposure of A375 cells to increasing concentrations of the compounds and reported as mean values ± s.d. from at least three independent experiments.

**Table 2 cells-08-01274-t002:** Stabilization of oncogene promoter G4-folded sequences (4 μM) by compound **1** (16 μM), measured by CD thermal unfolding analysis.

Oligonucleotide	T_m_ Values ^1^ (°C)			
	No Compound		c-exNDI 1	
	λ = 264 nm	λ = 290 nm	λ = 264 nm	λ = 290 nm
blc-2	67.5 ± 1.3	/	>90.0	/
c-myc	58.8 ± 0.7	/	>90.0	/
b-raf	>90.0	50.3 ± 0.3	>90.0	68.7 ± 0.8
c-kit 1	48.0 ± 0.3	/	>90.0	/
c-kit 2	57.0 ± 0.3	/	>90.0	/

^1^ Melting temperature of oncogene promoter G4-folded sequences in the absence or presence of a 4-fold excess of **1**. Data are reported as mean values ± sd from two independent experiments.

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
