# Peer review of "The Oncogenic Signaling Pathways in BRAF-Mutant Melanoma Cells Are Modulated by Naphthalene Diimide-Like G-Quadruplex Ligands"

_cells, 2019, doi:10.3390/cells8101274_

Round 1

Reviewer 1 Report

cells-569921

In this manuscript, the synthesis and screening of a library of differently functionalized core-extended naphthalene diimides for their activity against the BRAFV600E mutant melanoma cell line were performed. The biological effect was investigated in tumor cell lines and protein expression was examined by Western Blot analysis. The best compound was  able to stabilize G-quadruplexes that form in the promoter regions of KIT and BCL-2, target genes relevant to melanoma. This activity of the considered compound led to suppression of protein expression and thus to interference with oncogenic signaling pathways involved in BRAF-mutant melanoma cell survival, apoptosis and resistance to drugs.

The paper is a  high quality systematic study integrating biological and biophysical point of view. The biological implications are well articulated but the biophysical ones could be implemented. Maybe this implementation could perform also in a future work with the addition of an NMR study (in particular NOESY experiments) and molecular modeling, other that published by the same authors in J.Am.Chem.Soc. 2018, 140, 14528−1453.

Overall the work appears to be well conducted and contains the appropriate references with the following minor exception: the authors should add in the introduction more recent works as:

1) Loana Musso, Stefania Mazzini, Anna Rossini, Lorenzo Castagnoli, Leonardo Scaglioni, Roberto Artali, Massimo Di Nicola, Franco Zunino, Sabrina Dallavalle ”c-MYC G-quadruplex binding by the RNA polymerase I inhibitor BMH-21 and analogues revealed by a combined NMR and biochemical Approach” Biochimica et Biophysica Acta,  20181862, 615–629

https://doi.org/10.1016/j.bbagen.2017.12.002

2) Michael P. O'Hagan, Juan C. Morales and M. Carmen Galan “Binding and Beyond: What Else Can G-Quadruplex Ligands Do?” Eur. J. Org. Chem., 2019, DOI: 10.1002/ejoc.201900692

The problem and conclusions are clearly stated assuming that the G-quadruplex targeting is operative and that the c-KIT and BCL-2 protein level is modulated by several biochemical pathways.

The importance of the topic and the interesting results warrant publication in Cells.

Author Response

Point-by-point replies to Reviewers’ comments

Manuscript ref. cells-569921

The manuscript (ref. cells-569921) has been modified according to the useful observations and suggestions provided by the Reviewers. Enclosed please find the point-by-point replies to the Reviewers’ comments:

Reviewer #1

            In this manuscript, the synthesis and screening of a library of differently functionalized core-extended naphthalene diimides for their activity against the BRAFV600E mutant melanoma cell line were performed. The biological effect was investigated in tumor cell lines and protein expression was examined by Western Blot analysis. The best compound was able to stabilize G-quadruplexes that form in the promoter regions of KIT and BCL-2, target genes relevant to melanoma. This activity of the considered compound led to suppression of protein expression and thus to interference with oncogenic signaling pathways involved in BRAF-mutant melanoma cell survival, apoptosis and resistance to drugs.

The paper is a high quality systematic study integrating biological and biophysical point of view. The biological implications are well articulated but the biophysical ones could be implemented. Maybe this implementation could perform also in a future work with the addition of an NMR study (in particular NOESY experiments) and molecular modeling, other that published by the same authors in J.Am.Chem.Soc. 2018, 140, 14528−1453.

            Overall the work appears to be well conducted and contains the appropriate references with the following minor exception: the authors should add in the introduction more recent works as:

1) Loana Musso, Stefania Mazzini, Anna Rossini, Lorenzo Castagnoli, Leonardo Scaglioni, Roberto Artali, Massimo Di Nicola, Franco Zunino, Sabrina Dallavalle ”c-MYC G-quadruplex binding by the RNA polymerase I inhibitor BMH-21 and analogues revealed by a combined NMR and biochemical Approach” Biochimica et Biophysica Acta, 2018, 1862, 615–629 https://doi.org/10.1016/j.bbagen.2017.12.002

2) Michael P. O'Hagan, Juan C. Morales and M. Carmen Galan “Binding and Beyond: What Else Can G-Quadruplex Ligands Do?” Eur. J. Org. Chem., 2019, DOI: 10.1002/ejoc.201900692

            The problem and conclusions are clearly stated assuming that the G-quadruplex targeting is operative and that the c-KIT and BCL-2 protein level is modulated by several biochemical pathways.

The importance of the topic and the interesting results warrant publication in Cells.

Reply: We are grateful to the Reviewer for the extremely positive evaluation of our manuscript. We welcome the useful tips provided by the Reviewer. The proposed recent works has been added to the introduction section according to the Reviewer suggestions and are now references number 36 and 37. Changes made to the manuscript have been inserted using the Track Changes function in Microsoft Word.

Reviewer 2 Report

The present study aims to evaluate the anticancer properties of a series of NDI derivatives against melanoma. To this end, 14 derivatives were synthesized according to previously published procedures (17 PubMed entries with “Freccero” and “diimide”, 11 PubMed entries with “Richter” and “diimide”), providing a portfolio of derivatives bearing modifications classically seen in medchem research programs. Beyond this, this study aims to connect melanoma-responsible oncogenes (BRAF, Bcl2..) with quadruplexes (G4s) and G4-ligands, to assess whether G4-ligands might provide a novel therapeutic avenue for skin cancers. To this end, MTT assays are performed with the 14 derivatives to finally select the parent compound 1, which is unfortunate given that compound 1 has already been thoroughly investigated and will again be at the very heart of the present study. I suggest to move both Figure 1 and scheme 1 (redundant) along with the corresponding chemistry paragraph to the Supp. Info. given that they do not provide novelty. Beyond this, several remarks: A/ The down-regulation of transcripts involved in RAS/BRAF/MAPK and PI3K/AKT pathways along with BCL2 transcripts are interesting but must be confirmed by results obtained with a firmly the established G4 ligand pyridostatin (PDS) or, alternatively, BRACO19 or PhenDC (all of them are commercially available). This will lend further credence to the involvement of G4s in the modulation of the gene expression. B/ The discussion regarding the higher level of G4s found in rapidly dividing cells as compared to slowly dividing cells is rather speculative in its present state. I would suggest to remove it or the support it with firm experimental data (or suited bibliography). C/ melting experiments must be performed again with doubly-labeled oligos, given that FRET systems are far more sensitive than CD-melting. Experimental conditions (eg. potassium content) must be fine-tuned to obtain reliable (and comparable) T1/2 values. Duplex controls must be included. D/ The modulation of the protein levels observed after ligand treatment might also originate in a translational control involving G4 RNA. The authors must investigate the G4-RNA/compound 1 interactions and shed light on this mechanistic possibility. E/ The most important issue: it is intriguing that the DNA damaging capacity of these ligands is not more discussed in this manuscript, and even not at the very beginning of the manuscript. Indeed, it is now accepted that damage-associated genes are transcriptionally turned off, meaning that the effects observed here might originate in an alternative mechanism. This has to be thoroughly investigated and discussed given that, in its present state, this study fails in providing evidence of the G4 involvement in the observed global cellular effects. Do these damages involve G4? It might be possible but without answering this question, this manuscript raises more question than answers. Comparison with classical G4 binders (see above) and DNA damaging agents (eg. camptothecin) might help decipher the origins of the observed effects. For all these reasons, this manuscript is found too preliminary for being accepted for publication in Cells, but might be re-considered after substantial improvements.

Author Response

Point-by-point replies to Reviewers’ comments

Manuscript ref. cells-569921

The manuscript (ref. cells-569921) has been modified according to the useful observations and suggestions provided by the Reviewers. Enclosed please find the point-by-point replies to the Reviewers’ comments:

Reviewer #2

The present study aims to evaluate the anticancer properties of a series of NDI derivatives against melanoma. To this end, 14 derivatives were synthesized according to previously published procedures (17 PubMed entries with “Freccero” and “diimide”, 11 PubMed entries with “Richter” and “diimide”), providing a portfolio of derivatives bearing modifications classically seen in medchem research programs. Beyond this, this study aims to connect melanoma-responsible oncogenes (BRAF, Bcl2..) with quadruplexes (G4s) and G4-ligands, to assess whether G4-ligands might provide a novel therapeutic avenue for skin cancers. To this end, MTT assays are performed with the 14 derivatives to finally select the parent compound 1, which is unfortunate given that compound 1 has already been thoroughly investigated and will again be at the very heart of the present study.

We appreciated the observations raised by the Referee, which were very helpful to improve the scientific quality and readability of the manuscript. To address her/his issues, the main text has been edited. Changes made to the manuscript have been inserted using the Track Changes function in Microsoft Word.

I suggest to move both Figure 1 and scheme 1 (redundant) along with the corresponding chemistry paragraph to the Supp. Info. given that they do not provide novelty.

Reply: according to the reviewer’s request both Figure 1 and scheme 1 along with the corresponding chemistry paragraph have been moved to the “Supplementary Information” file.

Beyond this, several remarks:

A/ The down-regulation of transcripts involved in RAS/BRAF/MAPK and PI3K/AKT pathways along with BCL2 transcripts are interesting but must be confirmed by results obtained with a firmly the established G4 ligand pyridostatin (PDS) or, alternatively, BRACO19 or PhenDC (all of them are commercially available). This will lend further credence to the involvement of G4s in the modulation of the gene expression.

Reply: We would like to point out that in our study we did not claimed that modulation of transcripts involved in RAS/BRAF/MAPK and PI3K/AKT are for sure due only to the involvement of G4s. Instead, we clearly stated that the exposure to NDI resulted in a different outcome in terms of gene expression modulation in A375 vs. SKMEL-2 cells as well as that such an outcome may represent a distinct response of melanoma cells to the compound that may reflect their opposite genetic background (see paragraph 3.3, lines 345-349). In addition, we clearly stated that that the down-modulation of KIT protein levels was paralleled by the shutdown of each of MAPK and PI3K/AKT signaling pathways, as shown by the reduced levels of the phosphorylated forms of ERK1/2 and AKT (lines 452-456 and lines 521-524).

The emphasis we put on the NDI-mediated switch-off of MAPK and PI3K/AKT pathways was related to the important role that these pathways play in conferring resistance to targeted therapies in BRAF-mutant melanoma cells.

In addition, we also reported that the evidence that exposure of BRAF mutant melanoma cells to our G4 ligand resulted in the modulation in the expression levels of genes belonging to signaling paths that are relevant to melanoma cell survival and resistance to drug prompted us investigating whether the compound could interact with specific G4 targets at the interplay of RAS/RAF/MAPK and PI3K/AKT as well as of intrinsic apoptosis pathways. To this purpose, we focused on KIT, BRAF and BCL-2, owing both to their relevance as therapeutic targets in melanoma and to the evidence that they bear G4 forming sequences within the gene promoters (lines 360-365).

Obviously, as for any G4 ligand tested in cell cultures, we cannot exclude a priori that even the factors belonging to the MAPK and PI3K/AKT signaling pathways may represent possible G4 targets, among others, of NDI-mediated G4 stabilization as well as that the cell response to G4 ligands are largely dependent on the cellular context and on the presence of specific genetic alterations (see lines 534-537).

            It could be also questioned on whether or not the comparative assessment of the G4 stabilizing properties of compounds belonging to remarkably distinct chemical families, including the commercially available G4 ligands suggested by the Reviewer, may be scientifically accurate and informative. To the best of our knowledge, PDS and BRACO19 were all primarily tested as telomeric G4 ligands, and none of them have been reported thus far to specifically act at the promoter level of BCL-2, KIT, BRAF and/or MYC (Francisco AP, Paulo A. Curr Med Chem. 2017;24(42):4873-4904). The only exception is represented by a pyridostatin derivative reported by Feng Y. et al. (Bioorg Med Chem Lett. 2016 Apr 1;26(7):1660-3) who showed that the compound was able to inhibit BCL-2 transcriptional activation in just one in vitro model of head neck cancer (Hep-2 cell line). On the other hand, the seminal work by Rodriguez R et al. (Nat Chem Biol. 2012 Feb 5;8(3):301-10) showed that PDS produced transcription- and replication-dependent DNA damage both at telomeres and at several gene promoter sites in MRC-5-SV40 cells (i.e., SV40-infected human lung fibroblasts), in particular at a large G4 cluster in the promoter region of the SRC gene. In addition the authors explored whether PDS could affect the mRNA levels of MYC and of the top ten γH2AX-positive genes that contained the highest PQS densities identified by ChIP-Seq in PDS-treated MRC-5-SV40 cells. Though the authors did not give a precise indication on how the concentration of PDS used for this studies was selected, they reported that all the γH2AX-positive target genes analyzed, which included oncogenes (RHOC, FAIM2, SRC, VAV2, SREBF1, MYC) and tumor suppressor genes (CST6, LLGL1,NOTCH1, AHRR, C9orf140), were down-regulated after drug treatment, with the proto-oncogene SRC, but not MYC, being the most strongly affected. Moreover, in the same paper, the authors clearly showed that PDS did not particularly affect the in vitro growth of melanoma cells (8 cell line included in the study, see Supplementary figure S1 in Nat Chem Biol. 2012 Feb 5;8(3):301-10), thus highlighting that the compound is poorly active in melanoma cells.

To the best of our knowledge, there is no extensive analysis of the binding ability of BRACO-19 and PhenDC towards the promoters of KIT, BCL-2 or BRAF (Francisco AP, Paulo A. Curr Med Chem. 2017;24(42):4873-4904). We only found a couple of papers in literature, one related to NMR spectroscopy to solve the structure of the complex formed between Phen-DC3 and an intramolecular G-quadruplex derived from the c-myc promoter (Chung WJ. et al., Angew Chem Int Ed Engl. 2014 Jan 20;53(4):999-1002) and the other related to the 5'-UTR region of the TRF2 mRNA (Lopes J. et al., EMBO J. 2011 Aug 26;30(19):4033-46).

            Conversely, Prof. Stephen Neidle recently reported that the effects of NDI acting at promoter G4s may be multitarget. In particular, the tetrasubstituted compound (MM41) was examined in the MIA-PACA2 pancreatic cancer cell line and was shown to down-regulate the expression of several well-studied cancer genes, including BCL2 (see Nat Rev Chem. 1, 0041, 2017). Similarly, other NDIs, such as a tri-substituted naphthalene diimide, induced the downregulation of several genes in human melanoma and lung cancer cell lines, including telomerase reverse transcriptase (TERT) and BCL2 (Francisco AP, Paulo A. Curr Med Chem. 2017;24(42):4873-4904).

In addition, compound 1 is an established G4-ligand since it has been shown by our group and others that it specifically stabilizes G4 structures (see for example Perrone R et al., J Med Chem. 2015 Dec 24;58(24):9639-52; Zuffo M. et al., Biochim Biophys Acta Gen Subj. 2017 May;1861(5 Pt B):1303-1311; Zuffo M. et al., Nucleic Acids Res. 2018 Nov 2;46(19):e115; Platella C. et al., Anal Chim Acta. 2018 Nov 7;1030:133-141).

B/ The discussion regarding the higher level of G4s found in rapidly dividing cells as compared to slowly dividing cells is rather speculative in its present state. I would suggest to remove it or the support it with firm experimental data (or suited bibliography).

Reply: We understand the Reviewer’s skepticism regarding this specific aspect and we apologize for having formulated such a hypothesis, which of course is worth to be further investigated. Indeed, an aspect that is almost completely neglected thus far regards the lack of information on the kinetics of G4 generation, which in turn may depends on several factors (chromatin structure, superhelical stress, molecular crowding and the presence of G-quadruplex binding proteins, etc), as well as on G4 persistence during the natural history of tumor development and progression.

Anyway, to fulfill the reviewer’s request we removed this hypothesis and edited the text accordingly.

C/ melting experiments must be performed again with doubly-labeled oligos, given that FRET systems are far more sensitive than CD-melting. Experimental conditions (eg. potassium content) must be fine-tuned to obtain reliable (and comparable) T1/2 values. Duplex controls must be included.

Reply: As suggested by the Reviewer, FRET analysis was performed using a dual-labeled oligonucleotide at 100 mM and 10 mM of K+. We selected c-kit2 sequence as reference G4 forming oligonucleotide, and the melting profile was monitored by measuring FAM emission in the temperature range of 30-95 °C. A double stranded (ds) oligonucleotide was also used as a non-G4 forming sequence control to check 1 selectivity for G4 vs. duplex. In both K+ conditions, c-kit2 was greatly stabilized by 1 with Tm > 90 °C while the compound resulted basically devoid of stabilizing effect on dsDNA with variations in Tm values < 2 °C (see lines 392-400 and new Supporting Information Figure S4).

We would just like to point out that we deem the FRET experiments in general to be less straightforward as regard to data interpretation than CD analysis because of the possible modification of the G4 conformation induced by the fluorophores, as well as the possible interference of the ligands with the fluorophores. For this reason we initially presented only CD data, which are sensitive enough in our conditions to visualize the compound-dependent effect at the G4-folded oligonucleotide.

D/ The modulation of the protein levels observed after ligand treatment might also originate in a translational control involving G4 RNA. The authors must investigate the G4-RNA/compound 1 interactions and shed light on this mechanistic possibility.

Reply: We understand the interesting comment provided by the Reviewer, although his/her request sounds rather contradictory compared to that reported at point E/.

To investigate the possible involvement of G4-dependent inhibition of translation is of course an intriguing aspect to dig on at molecular level. However, in our opinion, the Reviewer’s request appears rather far beyond the aim of our study, which was not to investigate the action of a G4 ligand at a subtle mechanistic level, but rather to focus on sequences within the promoter of selected genes that are relevant for melanoma cells and found to be remarkably down-modulated at mRNA level in BRAF-mutant melanoma cells upon exposure to compound 1 (see lines 360-365).

We obviously cannot exclude a priori that the observed down-modulation of KIT and BCL-2 protein levels could in part originate from a G4-mediated translational control. However, providing biophysical data on a possible interaction between G4-RNA and compound 1 will not help uncovering its real involvement in the translational regulation of the target proteins at the cellular level and analyzing such interaction in living cells appears rather tricky presently.

Moreover, it should be taken into account that, except for the seminal paper by Rodriguez et al (Nat Chem Biol. 2012 Feb 5;8(3):301-10) showing that pyridostatin-induced γ-H2AX foci are enriched in quadruplex forming sequences (which does not mean that all G4 sequences are site of DNA damage in the presence of a given G4 ligand, see reply to point E/) to the best of our knowledge there are no published data where the G4 binding capability of a given ligand has been comprehensively investigated as a function of all possible quadruplex forming sequences within a cell system.

In addition, in the original version of the paper we did not claim anywhere that the observed effect are directly and solely due to the stabilization of G4 within the promoter of BCL2 and KIT genes, an aspect that held true for the supposed specificity of any G4 ligand and quadruplex forming sequence published thus far in literature (Francisco AP, Paulo A. Curr Med Chem. 2017;24(42):4873-4904). By contrast, we have clearly stressed that i) the observed effects may represent a distinct cell response to the compound that may reflect the opposite genetic background of the melanoma cell lines (see lines 345-349); ii) additional direct G4 targets (which may indeed include G4 RNA) may be amenable to 1-mediated G4 stabilization (lines 534-535) and iii) the cell response to G4 ligands are largely dependent on the cellular context (lines 536-537).

E/ The most important issue: it is intriguing that the DNA damaging capacity of these ligands is not more discussed in this manuscript, and even not at the very beginning of the manuscript. Indeed, it is now accepted that damage-associated genes are transcriptionally turned off, meaning that the effects observed here might originate in an alternative mechanism. This has to be thoroughly investigated and discussed given that, in its present state, this study fails in providing evidence of the G4 involvement in the observed global cellular effects. Do these damages involve G4? It might be possible but without answering this question, this manuscript raises more question than answers. Comparison with classical G4 binders (see above) and DNA damaging agents (eg. camptothecin) might help decipher the origins of the observed effects.

Reply: Unfortunately, we do not catch the Reviewer’s comment properly. As already stressed in the reply to comment D/ as well as in the original version of the paper, we did not claim anywhere that G4 involvement was the only cause of the observed global cellular effects. Conversely, in our opinion the observed phenotypes may likely fit with a mix of primary (G4-regulated) and secondary effects, including the cell response to stress induced by drug exposure. However, we have provided evidence that NDI exposure resulted in the induction of DNA damage, as revealed by the pronounced accumulation of the phosphorylated form of γ-H2AX, in A375 but not in SKMEL-2 cells (original Figure 4C now new Figure 3C and Figure S2). It is pretty intriguing that DNA damage induction was appreciable in cells characterized by a widespread down-modulation of gene expression and not in cells where only 10 genes out of 92 were found to be down-modulated upon NDI exposure (Supplementary Table S3). However, the evidence that damage-associated genes are transcriptionally turned off is not so straightforward. Indeed, in their seminal paper Rodiriguez R. et al. (Nat Chem Biol. 2012 Feb 5;8(3):301-10) identified genes containing quadruplex forming sequences (PQS) that were negative for γ-H2AX. For example, HRAS had a high PQS content but did not show detectable γ-H2AX enrichment in cells treated with pyridostatin. Thus, although there was a good correlation between PQS density and γH2AX formation for some genes, PQS density alone was not an accurate predictor of DNA damage induction and gene expression regulation through G4 targeting (Rodriguez et al., Nat Chem Biol. 2012 Feb 5;8(3):301-10). In addition, it has been reported that topotecan (i.e., a DNA damage inducing topoisomerase inhibitor) can recognize and bind a G-quadruplex formed within an upstream region from the transcription start site of c-Myb gene and favor the positive transcriptional regulation of the gene (Li F. et al., Int J Biol Macromol. 2018 Feb;107(Pt B):1474-1479), a mechanisms that is the opposite of the notion of G4-mediated turning-off of genes transcription in the presence of DNA damage. The same hold true for MYC the levels of which were found to be increased in A375 cells after a 72-h of NDI exposure (original Figure 4B, now new Figure 3B) as well as in lung cancer cells exposed to a different NDI derivative (Francisco AP, Paulo A. Curr Med Chem. 2017;24(42):4873-4904).

Anyway, in the revised version of the paper we now provide evidence that exposure of melanoma cells to the NDI derivative results in G4 folding, as detected by immunofluorescence analyses using the BG4 single chain antibody (see new Figure 1, panel C) as well as that treatment with NDI induced a marked induction of DNA damage, assessed in terms of γ–H2AX foci by immunofluorescence, in A375 cells and, to a very low extent, in SKMEL-2 cells (see new Supporting Information Figure S2A). In this context, a comparative analysis of G4 stabilization-mediated induction of DNA damage was also carried out in both melanoma cell lines upon exposure to the reference compound pyridostatin (see new Supporting information Figure S2A).

Finally, the comparison with classical G4 binders and DNA damaging agents (eg. camptothecin) to help deciphering the origins of the observed global cellular effects is likely poorly informative. Specifically, recent analysis of indenoisoquinoline (i.e., topoisomerase inhibitors) analogues for their MYC- and topoisomerase I-inhibitory properties reveals a synergistic effect of MYC and topoisomerase I inhibition on anticancer activity. This analysis has uncovered a novel mechanism of action of indenoisoquinolines as a new family of drugs targeting the MYC promoter G-quadruplex for MYC suppression and suggests that the dual targeting of MYC and topoisomerase I may serve as a novel strategy for anticancer drug development (Wang KB et al. J Am Chem Soc. 2019 Jul 17;141(28):11059-11070). Moreover, it has been demonstrated that telomeric G4 ligands cause an increase in the binding of Topoisomerase I at the telomeres and that the use of standard or novel camptothecins help stabilizing the G4 ligand-dependent DNA damage both in vitro and in xenografts (Biroccio A. et al. Clin Cancer Res. 2011 Apr 15;17(8):2227-36). Similarly, it has been reported that nemorubicin (a morpholinyl analogue of the anthracycline doxorubicin acting as a topoisomerase I inhibitor) and doxorubicin (an inhibitor of topoisomerase II), not only intercalate into the duplex DNA, but also result in the stabilization of G-quadruplex DNA within MYC promoter; which may in part explain the multiple mechanisms of action of their antitumor activity (Scaglioni L. et al. Biochim Biophys Acta. 2016 Jun;1860(6):1129-38). On the basis of these findings and on available literature data, comparing the DNA damage capacity of G4 ligands to that of “conventional” DNA damaging agents (e.g. camptothecins/anthracycline antibiotics) may lead to confounding and difficult-to-interpret results. Indeed, as shown for the anthracyclines, it cannot be excluded a priori that conventional DNA damaging agents have no effect at G-quadruplex levels.